# An LH1–RC photocomplex from an extremophilic phototroph provides insight into origins of two photosynthesis proteins

Kazutoshi Tani [1,9,10✉], Ryo Kanno[2,9], Keigo Kurosawa[3], Shinichi Takaichi [4], Kenji V. P. Nagashima[5], Malgorzata Hall[2], Long-Jiang Yu [6], Yukihiro Kimura [7], Michael T. Madigan[8], Akira Mizoguchi[1], Bruno M. Humbel[2] & Zheng-Yu Wang-Otomo [3,10✉]

*Rhodopila globiformis* is the most acidophilic of anaerobic purple phototrophs, growing optimally in culture at pH 5. Here we present a cryo-EM structure of the light-harvesting 1–reaction center (LH1–RC) complex from *Rhodopila globiformis* at 2.24 Å resolution. All purple bacterial cytochrome (Cyt, encoded by the gene *pufC*) subunit-associated RCs with known structures have their N-termini truncated. By contrast, the *Rhodopila globiformis* RC contains a full-length tetra-heme Cyt with its N-terminus embedded in the membrane forming an α-helix as the membrane anchor. Comparison of the N-terminal regions of the Cyt with PufX polypeptides widely distributed in *Rhodobacter* species reveals significant structural similarities, supporting a longstanding hypothesis that PufX is phylogenetically related to the N-terminus of the RC-bound Cyt subunit and that a common ancestor of phototrophic *Proteobacteria* contained a full-length tetra-heme Cyt subunit that evolved independently through partial deletions of its *pufC* gene. Eleven copies of a novel γ-like polypeptide were also identified in the bacteriochlorophyll *a*-containing *Rhodopila globiformis* LH1 complex; γ-polypeptides have previously been found only in the LH1 of bacteriochlorophyll *b*-containing species. These features are discussed in relation to their predicted functions of stabilizing the LH1 structure and regulating quinone transport under the warm acidic conditions.

[1] Graduate School of Medicine, Mie University, Tsu 514-8507, Japan. [2] Imaging Section, Research Support Division, Okinawa Institute of Science and Technology Graduate University (OIST), 1919-1, TanchaOnna-son Kunigami-gun, Okinawa 904-0495, Japan. [3] Faculty of Science, Ibaraki University, Mito 310-8512, Japan. [4] Department of Molecular Microbiology, Faculty of Science, Tokyo University of Agriculture, Sakuragaoka, Setagaya Tokyo 156-8502, Japan. [5] Research Institute for Integrated Science, Kanagawa University, 2946 Tsuchiya, Hiratsuka, Kanagawa 259-1293, Japan. [6] Photosynthesis Research Center, Key Laboratory of Photobiology, Institute of Botany, Chinese Academy of Sciences, 100093 Beijing, China. [7] Department of Agrobioscience, Graduate School of Agriculture, Kobe University, Nada Kobe 657-8501, Japan. [8] School of Biological Sciences, Department of Microbiology, Southern Illinois University, Carbondale, IL 62901, USA. [9] These authors contributed equally: Kazutoshi Tani, Ryo Kanno. [10] These authors jointly supervised this work: Kazutoshi Tani, Zheng-Yu Wang-Otomo. ✉email: ktani@doc.medic.mie-u.ac.jp; wang@ml.ibaraki.ac.jp

The photooxidized bacteriochlorophyll (BChl) dimer (special pair) in the photosynthetic reaction center (RC) of anoxygenic phototrophic bacteria is reduced either directly by a soluble redox protein or indirectly through a membrane-associated heme protein interpositioned between the soluble components and the photooxidized special pair, or by both in some species. Soluble electron donors are typically a small heme protein or a high-potential iron-sulfur protein, while the electron mediating membrane proteins include the RC-bound multi-heme cytochrome (Cyt) in many purple bacteria[1] and in green filamentous bacteria[2,3], or the multiple copies of a membrane-anchored mono-heme Cyt found in some purple bacteria[4] and also in green sulfur bacteria[5,6] and the heliobacteria[5,7]. Electron mediation through a membrane-associated Cyt is most common[1] and is thought to be a more ancient form of photosynthesis[1,5,8].

In purple phototrophs, the RC-bound multi-heme Cyt subunits are classified into two groups based on whether or not a lipoprotein signal peptide followed by a cysteine (Cys) residue exists in its N-terminus[9,10]. The Cys-containing Cyt subunits with known structures all have their N-termini posttranslationally truncated (about 20 residues removed) at this Cys residue and are then modified by a covalently bound tri- or diacylglycerol lipid; such is characteristic of *Blastochloris* (*Blc.*) *viridis*[11,12], *Thermochromatium* (*Tch.*) *tepidum*[13], *Thiorhodovibrio* (*Trv.*) strain 970[14], and *Allochromatium* (*Alc.*) *tepidum*[15]. Since the N-terminus-truncated Cyt subunits have no transmembrane domain, they are anchored by the covalently bound lipid in the membrane-embedded RC. By contrast, a Cys-lacking Cyt subunit from the aerobic purple bacterium *Roseobacter* (*Rsb.*) *denitrificans* has a full N-terminus unmodified in the purified RC[10], and based on its sequence, presumably forms an α-helix inserted in the membrane as an anchor instead of lipid[16]. Predictions based on sequence analyses also suggest that this may be the case in *Rhodovulum* (*Rdv.*) *sulfidophilum*, *Acidiphilium* (*Acp.*) *rubrum*, and *Rhodospirillum* (*Rsp.*) *molischianum*[10].

In contrast to these, *Rhodobacter* (*Rba.*) species lack an RC-bound Cyt subunit and the gene that encodes it (*pufC*) in their *puf* operon. Instead, these species use both a soluble Cyt $c_2$[17] and a membrane-anchored mono-heme Cyt $c_y$[4] to facilitate electron transfer to the RC. In all *Rhodobacter* species, there is a unique protein in the light-harvesting 1–reaction center (LH1–RC) complex called PufX that interacts with the RC-L subunit and LH1 polypeptides[18–23]. PufX (encoded by the gene *pufX*[24,25]) is a transmembrane protein composed of about 80 amino acids, and *pufX* is located at the same position as *pufC*—immediately following *pufM* (the gene encoding the RC-M subunit) in the *puf* operon. PufX plays important roles in the regulation of cyclic electron transfer and photosynthetic membrane morphology and core complex organization[26], and its transmembrane portion has structural motifs similar to the N-terminal (and likely transmembrane) helix of the *Rsb. denitrificans* Cyt subunit[10,16]. These findings have led to a hypothesis that PufX has phylogenetic roots in the N-terminal domain of the RC-bound Cyt subunit[10].

Here, we lend experimental support for this sequence-based hypothesis with structural data from a cryo-EM analysis of the LH1–RC complex from the phylogenetically and physiologically unique purple phototrophic bacterium *Rhodopila* (*Rpi.*) *globiformis*[27]. This phototroph inhabits warm acidic springs at pH values as low as 3 and is the most acidophilic of all known anaerobic purple phototrophs[28]. The *Rpi. globiformis* RC complex contains a full-length Cyt subunit whose structure confirms that its N-terminal region forms a transmembrane helix within the LH1–RC complex. The N-terminal portion of the Cyt subunit also displays a marked similarity to PufX, providing crucial new insights into the relationship between the two proteins. Moreover, the structure of the bacteriochlorophyll (BChl) *a*-containing *Rpi.*

*globiformis* LH1 complex revealed multiple copies of a polypeptide absent from the LH1 complexes of other BChl *a*-containing species but related to the γ-polypeptides in the LH1 complexes of BChl *b*-containing purple bacteria. Collectively, these data show the *Rpi. globiformis* LH1–RC complex to be structurally unique among purple bacteria and provide experimental evidence of a phylogenetic link between PufX and the RC Cyt subunit.

## Results

**Structural overview of the *Rpi. globiformis* LH1–RC.** Purified LH1–RC from cells of *Rpi. globiformis* DSM161[T] exhibited an LH1 absorption maximum at 879 nm (Supplementary Fig. 1). The cryo-EM structure of the core complex was determined at 2.24 Å resolution (Table 1 and Supplementary Figs. 2–4). The LH1 complex forms a closed ring structure (Fig. 1a, b) with an arrangement of the α- and β-polypeptides similar to that of purple bacteria such as *Tch. tepidum*[13], *Trv.* strain 970[14], *Rhodospirillum rubrum*[29,30] and *Alc. tepidum*[15]. In addition to the 16 pairs of αβ-polypeptides, several fragments of electron potential densities with similar length and α-helical characteristic were aligned in the transmembrane region between β-polypeptides in the refined map. These were assumed to be the same protein but could not be modeled by any annotated protein sequences.

**Table 1 Cryo-EM data collection, refinement, and validation statistics.**

|  | LH1-RC-complex (EMDB-33501, PDB ID: 7XXF) |
| --- | --- |
| *Data collection and processing* | |
| Magnification | 96,000 |
| Voltage (kV) | 300 |
| Electron exposure (e−/Å²) | 40 |
| Defocus range (μm) | −0.6 to −2.8 |
| Pixel size (Å) | 0.820 |
| Symmetry imposed | C1 |
| Initial particle images (no.) | 421,464 |
| Final particle images (no.) | 128,119 |
| Map resolution (Å) | 2.2 |
| FSC threshold | 0.143 |
| Map resolution range (Å) | 313–2.2 |
| *Refinement* | |
| Initial model used (PDB code) | 5Y5S |
| Model resolution (Å) | 2.3 |
| FSC threshold | 0.5 |
| Model resolution range (Å) | 142–2.2 |
| Map sharpening *B* factor (Å²) | −40 |
| Model composition | |
| Non-hydrogen atoms | 29,826 |
| Protein residues | 3004 |
| Ligands | 98 |
| Water | 538 |
| *B* factors (Å²) | |
| Protein | 38.0 |
| Ligand | 42.4 |
| Water | 40.8 |
| R.m.s. deviations | |
| Bond lengths (Å) | 0.006 |
| Bond angles (°) | 2.574 |
| Validation | |
| MolProbity score | 1.65 |
| Clashscore | 7.30 |
| Poor rotamers (%) | 1.31 |
| Ramachandran plot | |
| Favored (%) | 97.1 |
| Allowed (%) | 2.9 |
| Disallowed (%) | 0.0 |

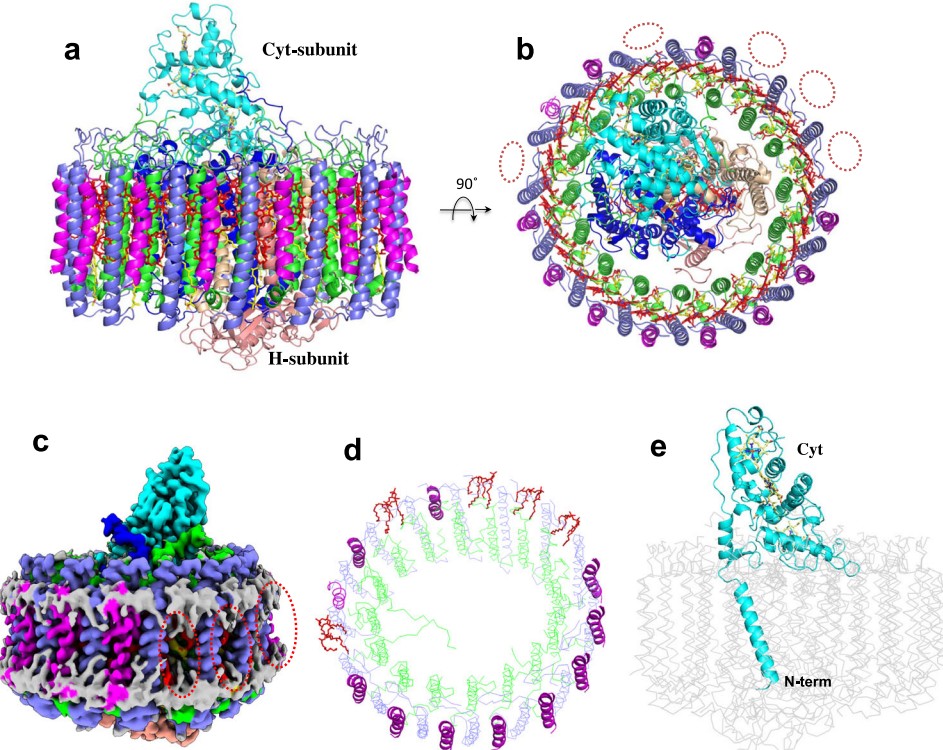

**Fig. 1 Structure overview of the LH1–RC core complex from *Rpi. globiformis*. a** Side view of the core complex parallel to the membrane plane. **b** Top view from the periplasmic side of the membrane. Dotted ovals indicate the positions that lack the γ-like polypeptide. **c** Side view of surface representation of the complex. Dotted red ovals indicate the positions that lack the γ-like polypeptide. **d** Tilted view showing distributions of the γ-like polypeptides (magenta cartoons) and cardiolipins (red sticks) in the LH1 ring. **e** Side view of the Cyt subunit (cyan) in the LH1–RC showing its N-terminal transmembrane domain. Heme groups are shown by paleyellow sticks. Color scheme: LH1-α, green; LH1-β, slate blue; LH1-γ, magenta; RC-C, cyan; RC-L, wheat; RC-M, blue; BChl *a*, red sticks; carotenoids, yellow sticks; BPhe *a*, magenta sticks; lipids or detergents, gray.

However, because of the high quality of our density map, we were able to trace most residues of this unknown protein. The derived amino acid sequence allowed us to search the *Rpi. globiformis* DSM 161[T] genome[31] using TBLASTN[32], and in so doing, a 69 bp nucleotide sequence encoding 22 amino acids (NCBI Reference Sequence: NZ_NHRY01000272.1) (Supplementary Fig. 5) emerged whose translated amino acid sequence precisely fit the density map (Supplementary Fig. 4b). Hereafter, we refer to this previously unannotated protein as "γ-like polypeptide", a structural analogue to the LH1 γ-polypeptides found thus far only in BChl *b*-containing LH1 complexes[33–36]; a total of 11 γ-like polypeptides were present per *Rpi. globiformis* LH1 complex (Fig. 1c, d).

The *Rpi. globiformis* RC complex also contained a highly unusual Cyt subunit with its full N-terminal domain present and embedded in the transmembrane region forming an α-helix as the membrane anchor (Fig. 1e). This arrangement differs from that in all other purple bacterial RC-bound Cyt subunits with known structures; in the latter, the N-termini are truncated and modified by acyl lipids that function as membrane anchors. The intact (untruncated) Cyt structure in *Rpi. globiformis* may therefore be an ancestral form of the RC-bound Cyt from which more compact (truncated) versions evolved.

The *Rpi. globiformis* LH1–RC contains 36 molecules of BChls *a* (32 in the LH1 and 4 in the RC) (Supplementary Fig. 15, Supplementary Table 1) and 17 keto-carotenoids (16 *trans*-carotenoids in LH1 and 1 *cis*-carotenoid in RC), the latter pigments are unique to this species[27,37]. Four types of keto-carotenoids were reported in the *Rpi. globiformis* membranes[37], and were confirmed in our work (Supplementary Fig. 6). Because

the R.g. keto-II type carotenoid is dominant in the purified LH1–RC (Supplementary Fig. 6), it was modeled in the LH1 structure with its keto group located on the periplasmic side. Three types of quinones, ubiquinone (UQ), menaquinone (MQ) and rhodoquinone, are present in *Rpi. globiformis* membranes, with UQ predominating (Supplementary Fig. 7). Based on the density map, one MQ and one UQ were identified at the $Q_A$ and $Q_B$ sites in the RC (Supplementary Fig. 4b, Supplementary Fig. 15), respectively. Electron potential density corresponding to a disulfide bond (Cys170–Cys172) in the RC-M subunit on the periplasmic surface is clearly visible in Supplementary Fig. 4b, indicating that our cryo-EM map was free from electron damage[38].

**The Cyt subunit in the *Rpi. globiformis* RC.** The *Rpi. globiformis* Cyt subunit contains fewer charged residues (30 and 24 of basic and acidic residues, respectively) than that of the neutrophilic *Blc. viridis* (39 and 28 of basic and acidic residues, respectively), a tendency also displayed by the extremely acidophilic aerobic purple bacterium *Acp. rubrum*[39]. Moreover, the surface of the membrane-extruded portion of the *Rpi. globiformis* Cyt subunit is slightly more basic than the Cyt subunits of *Blc. viridis* and *Tch. tepidum* (Fig. 2a, Supplementary Fig. 8). This is likely a function of the acidophilic nature of *Rpi. globiformis*[27] and is consistent with the observation that lowering pH results in a relatively more basic surface of the RC-bound Cyt subunit[40]. The purified *Rpi. globiformis* LH1–RC also showed some degree of thermostability over a pH range of 5.0–7.5 and a slightly lower stability at higher pH (Supplementary Fig. 8c); in nature, *Rpi. globiformis* inhabits

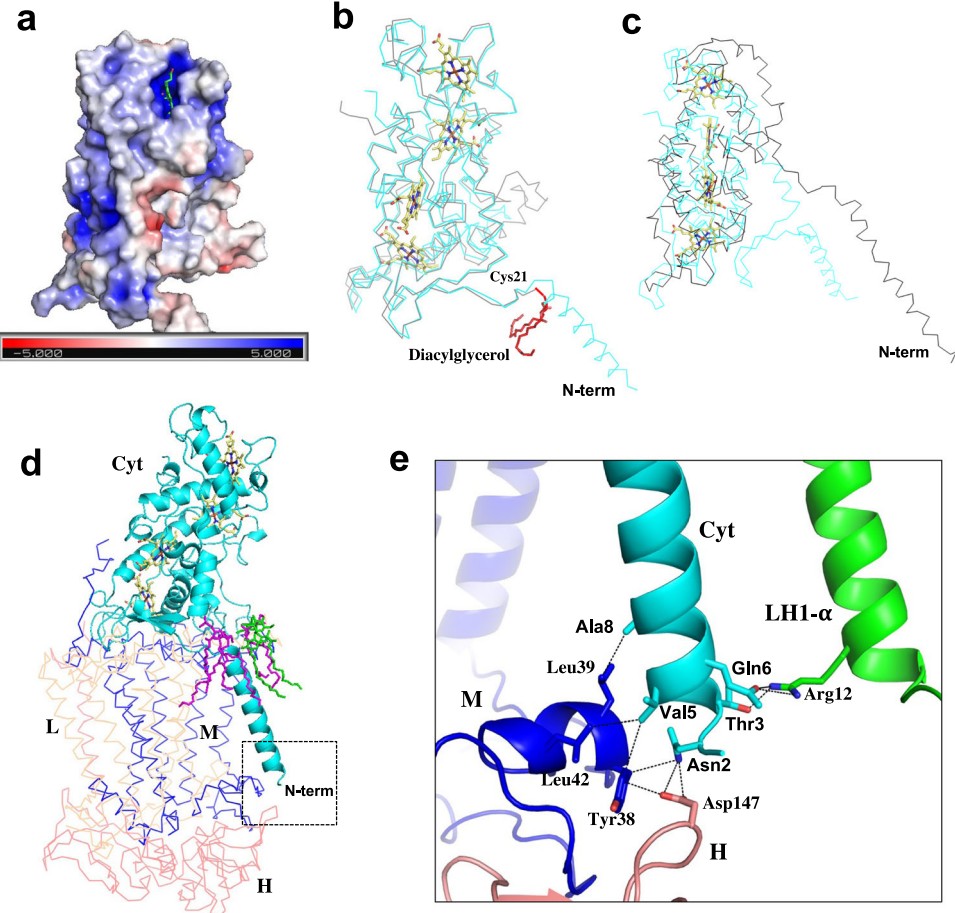

**Fig. 2 Structure of the Cyt subunit in the *Rpi. globiformis* RC. a** Surface charge distributions of the Cyt subunit with color codes according to the electrostatic potential from −5.0 $k_B$T (red, negative charge) to +5.0 $k_B$T (blue, positive charge) in the scale bars. Heme-1 is shown by green sticks. **b** Superposition of the Cα carbons of the Cyt subunits between *Rpi. globiformis* (cyan) and *Blc. viridis* (PDB: 3T6E, gray). The diacylglycerol group attached to the N-terminal Cys21 in *Blc. viridis* is shown by red sticks. Heme groups in the *Rpi. globiformis* Cyt subunit is shown by pale yellow sticks. **c** Superposition of the Cα carbons of the Cyt subunits of *Rpi. globiformis* (cyan) and *Rof. castenholzii* (PDB: 5YQ7, black). **d** The Cyt subunit (cyan cartoon) in the *Rpi. globiformis* RC showing its N-terminal domain surrounded by phosphatidylglycerol (magenta sticks) and detergent (DDM, green sticks) molecules on the periplasmic surface. **e** Expanded view of the N-terminus of the Cyt subunit marked in **d** showing interactions with the surrounding M subunit (blue), H-subunit (salmon) and LH1 α-polypeptide (green). Dashed lines indicate several representative distances shorter than 4.0 Å.

warm acidic springs near 40 °C[27,28]. The overall structure of the *Rpi. globiformis* Cyt subunit including the four heme groups is similar to that of the *Blc. viridis*[12,41] except, and importantly, for the N-terminal domains (Fig. 2b). The N-terminus of the *Rpi. globiformis* Cyt subunit is extended into the transmembrane region and forms an α-helix anchor in the membrane; by contrast, the *Blc. viridis* Cyt subunit is truncated at Cys21 and contains a diacylglycerol as the membrane anchor[11]. The latter structure is also found in the RC-bound Cyt subunits from *Tch. tepidum*[13], *Trv.* strain 970[14] and *Alc. tepidum*[15]. The transmembrane domain of the *Rpi. globiformis* Cyt subunit is surrounded by phospholipids and detergents on the periplasmic surface (Fig. 2d), implying a propensity of this portion for lipid molecules. The N-terminus of the Cyt subunit from Asn2 to Ala8 interacts with the RC-M and RC-H proteins and a nearby LH1 α-polypeptide on the cytoplasmic surface (Fig. 2e), mainly through charge/polar and hydrogen bond interactions. In summary, the extensive interactions of the *Rpi. globiformis* Cyt subunit transmembrane N-terminal domain with surrounding proteins and lipids not only make the subunit structurally unique among anaerobic purple bacteria but firmly anchor the Cyt subunit and thus stabilize the entire LH1–RC complex.

**Comparison of the *Rpi. globiformis* Cyt subunit with PufX in *Rhodobacter* species.** Because the structure of the *Rpi. globiformis* LH1–RC determined here is the first in an anaerobic purple bacterium containing an intact RC-bound Cyt with a full N-terminus, we were able to test a hypothesis surrounding the relationship between the N-terminal portion of the intact Cyt and the PufX polypeptide in the LH1–RC from *Rhodobacter* species (Supplementary Fig. 9)[10]. Superposition of the RC-M proteins from *Rpi. globiformis* and *Rba. sphaeroides* revealed that the C-terminus (Leu55–Gly69) of PufX is indeed closely located and parallel to a loop region (Phe27–Arg40) of the Cyt subunit immediately following its N-terminal transmembrane domain (Fig. 3a, b, f). The positional and conformational overlaps between the two fragments along with a similar topology of their transmembrane helices are structurally remarkable features and support the contention that PufX from *Rhodobacter* species may have phylogenetic roots in the intact N-terminus of the Cyt subunit.

Further evidence for a structural relationship between the C-terminus of PufX and the N-terminus of the RC-bound Cyt subunit can be observed for the Cyt subunits with truncated N-termini (Fig. 3, Supplementary Fig. 10). The N-terminal

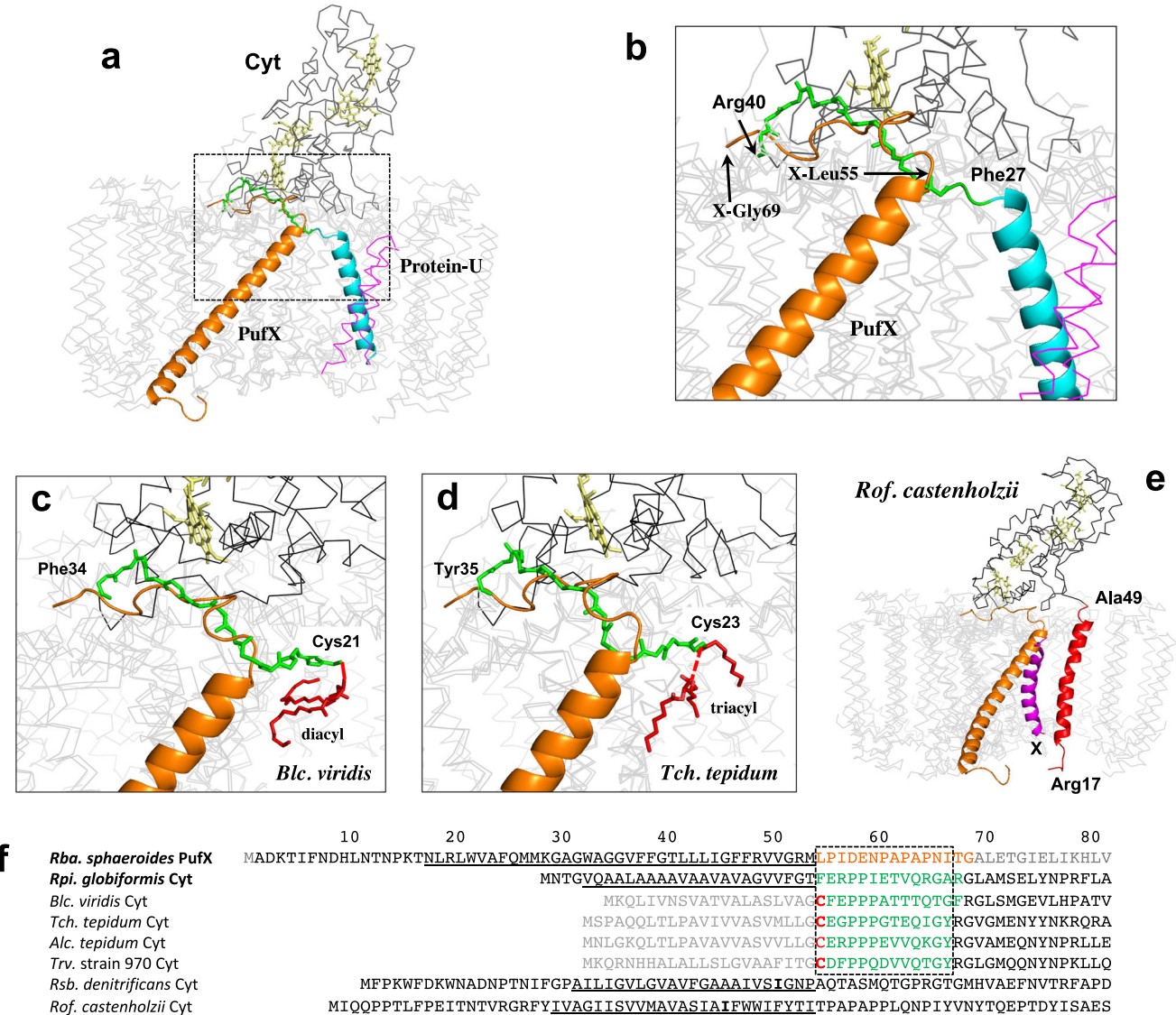

**Fig. 3 Structural similarities between PufX and the N-termini of the RC-bound Cyt subunits. a** Superposition of the Cα carbons of the RC-M subunits between the *Rba. sphaeroides* monomeric LH1–RC (PDB: 7F0L) and *Rpi. globiformis* RC (this work). PufX is shown as orange cartoon. The transmembrane helix and the following loop region of the Cyt subunit are shown by cyan cartoon and green sticks (main chain only), respectively. The other portion of the Cyt subunit is shown by a black ribbon and heme groups are shown by pale yellow sticks. Protein-U in *Rba. sphaeroides* LH1–RC is shown by magenta ribbon. **b** Expanded view marked in **a** showing that the C-terminus (Leu55–Gly69) of PufX overlaps positionally and conformationally with the N-terminal loop region (Phe27–Arg40, colored green) of the *Rpi. globiformis* Cyt subunit. **c** Similar comparison between the C-terminus of *Rba. sphaeroides* PufX and the N-terminus (Cys21–Phe34, colored green) of *Blc. viridis* Cyt subunit (PDB: 3T6E). **d** Similar comparison between the C-terminus of *Rba. sphaeroides* PufX and the N-terminus (Cys23–Tyr35, colored green) of *Tch. tepidum* Cyt subunit (PDB: 5Y5S). **e** Comparison of the *Rba. sphaeroides* PufX with the N-terminal transmembrane domain (Arg17–Ala49, red cartoon) of *Rof. castenholzii* Cyt subunit in an LH–RC structure (PDB: 5YQ7). A hypothetical subunit X is shown by magenta cartoon. **f** Sequence alignments of *Rba. sphaeroides* PufX (from strain IL106) with the N-termini of the Cyt subunits from other phototrophs based on the residues in the loop regions marked by dashed box or the transmembrane domains. Residues in the C-terminus (Leu55–Gly69) of PufX are shown by orange fonts. Residues truncated by posttranslational modification are shown by gray fonts. The Cys residues that bind a triacyl or diacyl group are shown by red fonts. Residues with green fonts correspond to the N-terminal portions shown in **a–d**. Transmembrane regions are underlined.

stretches of Cyt subunits containing about 14 residues after posttranslational truncation from *Blc. viridis*, *Tch. tepidum*, *Trv.* strain 970 and *Alc. tepidum* all closely overlap with the C-terminal domain (Leu55–Gly69) of PufX on the periplasmic surface, indicating that these portions are conserved in both proteins (Fig. 3c, d, f, Supplementary Fig. 10). It is of interest to note that an RC-bound Cyt subunit from the phylogenetically distinct ancient filamentous anoxygenic phototroph *Roseiflexus* (*Rof.*) *castenholzii* also has an intact N-terminal domain[3] (Fig. 2c), but its N-terminal portion (transmembrane helix and the following loop region) is located at a slightly different position

from that of PufX (Fig. 3e). An additional hypothetical subunit X (63 residues) in *Rof. castenholzii* was found near to where PufX is positioned in *Rhodobacter* species[3].

**Anchoring mechanism of the *Rpi. globiformis* Cyt subunit to the LH1–RC.** In addition to anchoring the *Rpi. globiformis* Cyt subunit by its N-terminal transmembrane domain, four LH1 α-polypeptides tightly associate the soluble portion of the Cyt subunit with their C-termini in a manner similar to that seen with the α3-polypeptide (the longest chain) in the *Alc. tepidum* LH1[15] and the α2- and α4-polypeptides (also the longest chains) in the

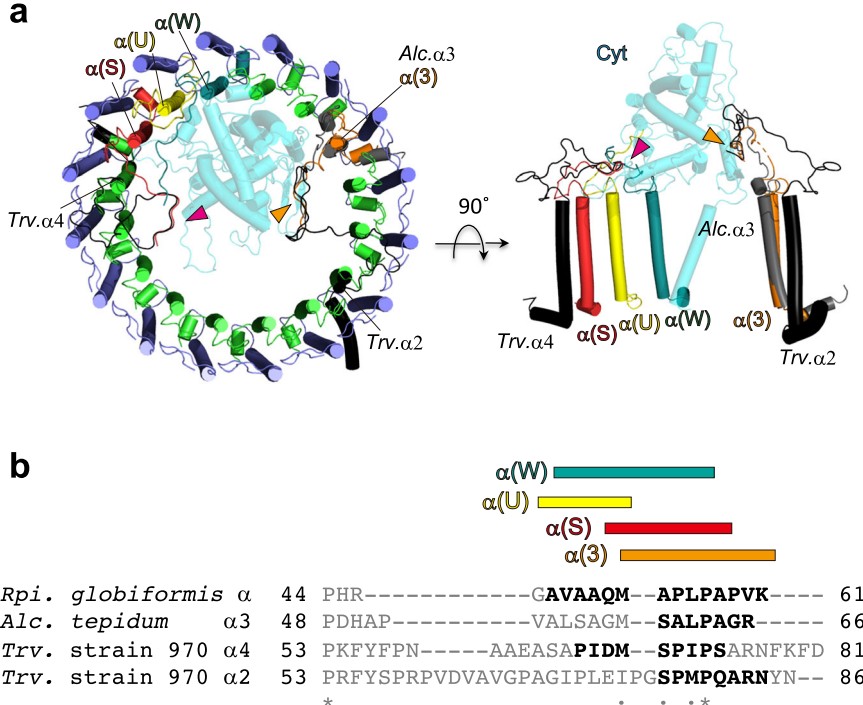

**Fig. 4 Interactions between the Cyt subunit and LH1 α-polypeptides. a** Left panel shows a top view of interactions from the periplasmic side between the *Rpi. globiformis* α-polypeptides (chain 3, orange; chain S, red; chain U, yellow; chain W, turquoise blue) and the Cyt subunit (transparent cyan); arrow heads indicate the overlapping C-terminal regions between the *Rpi. globiformis* α-polypeptides and the *Alc. tepidum* α3-polypeptide (PDB: 7VRJ, gray) and the α2- or α4-polypeptide of *Trv.* strain 970 (PDB: 7C9R, black). Other color scheme: LH1-α, green; LH1-β, slate blue. Right panel is a side view showing the four *Rpi. globiformis* α-polypeptides (colored) interacting with the Cyt subunit. For comparison, the *Alc. tepidum* α3-polypeptide and the α2- or α4-polypeptide of *Trv.* strain 970 are shown by gray and black cartoons, respectively. **b** Alignment of the C-terminal amino acid sequences of the α-polypeptides between *Rpi. globiformis*, *Alc. tepidum* and *Trv.* strain 970. Bold letters indicate residues in the overlapping regions of the structures. Symbol scheme: identical (*), conservative (:), semi-conservative (.). Upper colored bars represent regions of the *Rpi. globiformis* α-polypeptides that interact with the Cyt subunit. The color code is the same as in **a**.

*Trv.* strain 970 LH1[14] (Fig. 4a). In fact, one *Rpi. globiformis* α-polypeptide (chain ID:3) structurally overlaps with the *Alc. tepidum* α3-polypeptide and the *Trv.* strain 970 α2-polypeptide at their C-termini, and another (chain ID: S) overlaps with the *Trv.* strain 970 α4-polypeptide. Although overall sequence comparison of the *Rpi. globiformis* α-polypeptide with these polypeptides shows relatively low similarities for their C-termini, local alignment reveals a key motif [A/S]-[P/A]-[L/M/I]-P (Fig. 4b, Supplementary Fig. 11) at their distal C-terminal ends that is likely responsible for the extensive interactions with their corresponding Cyt subunits. The *Rpi. globiformis* Cyt subunit also has a larger surface contact area (332.1 Å²) with its LH1–RC than do the other LH1–RC (159.6 Å² for *Trv.* strain 970 and 138.4 Å² for *Alc. tepidum*). In addition, despite low sequence similarities, the RC-M subunits interact extensively with the Cyt subunits at their C-termini with similar structural features for all Cyt subunit-containing species (Supplementary Fig. 12). These data strongly suggest that the interacting residues between the two subunits evolved simultaneously to form suitable surface recognition contacts.

**The novel γ-like polypeptides in the *Rpi. globiformis* LH1 complex**. The newly identified γ-like polypeptide from our cryo-EM density map of the purified *Rpi. globiformis* LH1–RC was confirmed by mass spectroscopy; the novel polypeptide contained 22 residues with its N-terminal Met modified by a formyl group (Supplementary Fig. 13). The amino acid sequence exhibited high hydrophobicity, in good agreement with prediction by a membrane protein topology analysis[42] that the γ-like polypeptide is an

integral membrane protein (Supplementary Fig. 5c). Eleven γ-like polypeptides are located between β-polypeptides in the LH1–RC with an opposite orientation (N-terminus on the periplasmic side) to the α- and β-polypeptides (Fig. 5a); the five remaining sites were occupied by cardiolipin molecules (Fig. 1d, Fig. 5b). Inspection of the asymmetrically distributed binding sites of the γ-like polypeptides revealed a high structural homogeneity in their binding environments with nearby β-polypeptides (Supplementary Fig. 14b). By contrast, the cardiolipin-binding sites exhibited relatively large structural inhomogeneities (Supplementary Fig. 14b), indicating specific bindings of the γ-like polypeptide and cardiolipin to these sites.

The γ-like polypeptides interact over the entire range with neighboring β-polypeptides, mainly through hydrophobic interactions (Fig. 5c), enhancing stability of the LH1 ring. Moreover, the γ-like polypeptides block 11 of the 16 pores formed between LH1-αβ pairs that presumably function as quinone transport channels (Fig. 5d); one other channel is blocked by an additional UQ and a carotenoid (Supplementary Fig. 15b, c) leaving four channels in an open state. This incomplete blockage of the circular LH1 "fence" may be a mechanism for regulating the rate of quinone transport (Supplementary Fig. 16). Also, because the one channel blocked by UQ and a carotenoid is near the $Q_A$ site, the route for quinone access is limited to the $Q_B$ site. Such a partial blockage arrangement implies a mechanism for quinone exchange only through the $Q_B$ site.

The γ-like polypeptide in the *Rpi. globiformis* LH1 complex is the first (and thus far only) example of such a protein in BChl *a*-containing purple phototrophs. The polypeptides are

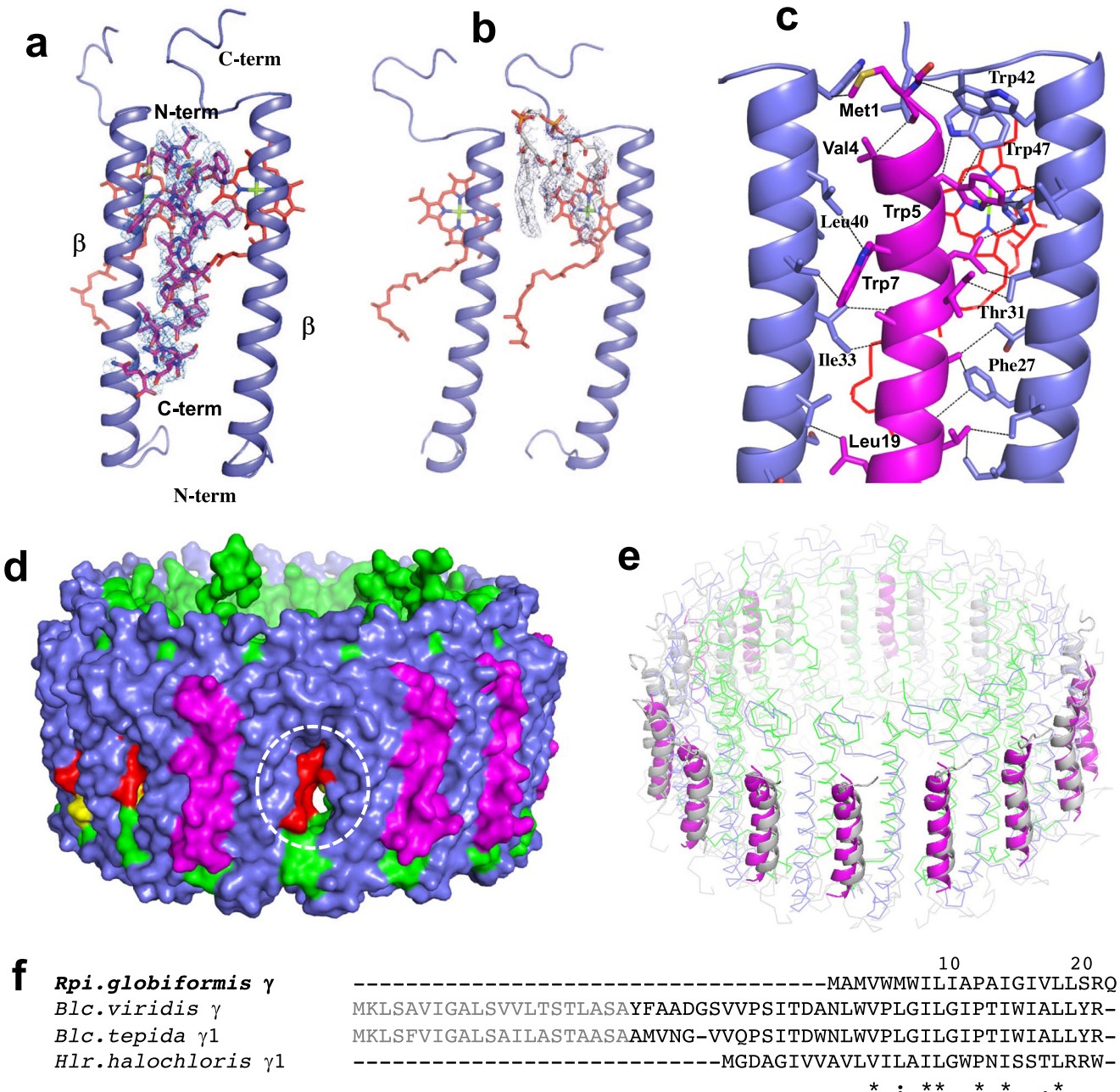

**Fig. 5 Structure of the γ-like polypeptides in the *Rpi. globiformis* LH1. a** Density map of a typical γ-like polypeptide with its nearby LH1 β-polypeptides showing the opposite orientations. The density map is shown at a contour level of 4.0 σ. **b** Density map of a cardiolipin with its nearby LH1 β-polypeptides at the location that lacks a γ-like polypeptide. **c** Interactions between a γ-like polypeptide (magenta) with its neighboring LH1 β-polypeptides (slate blue) with the dashed lines representing typical close contacts (<4.0 Å) between the two polypeptides. A BChl *a* molecule is shown by red sticks. **d** Side view of a surface representation showing a channel (white dashed circle) in the LH1 ring which lacks a γ-like polypeptide, other channels are sealed by the γ-like polypeptides (magenta). Color scheme: LH1-α, green; LH1-β, slate blue; BChl *a*, red; carotenoids, yellow. **e** Superposition of the LH1 polypeptides between *Rpi. globiformis* (colored) and *Blc. viridis* (PDB: 6ET5, gray) showing the structural similarity between the γ-polypeptides (cartoon). **f** Alignment of the γ-polypeptides between *Rpi. globiformis* and BChl *b*-containing species with identical amino acids (*), conservative (:) and semi-conservative (.). The N-terminal fragments in *Blastochloris* species shown by gray fonts were not identified in the purified protein. *Hlr*: Halorhodospira.

positioned and show a similar conformation and orientation as do the LH1 γ-polypeptides in the BChl *b*-containing bacterium *Blc. viridis*[35] (Fig. 5e). However, differing from *Blc. viridis*, the *Rpi. globiformis* γ-like polypeptides do not interact with LH1 α-polypeptides because the *Rpi. globiformis* α-polypeptide has a much shorter N-terminus than that of *Blc. viridis*. Although comparison of amino acid sequences over the transmembrane domain revealed that the full-length *Rpi. globiformis* γ-polypeptide has relatively low sequence similarities to those of the expressed γ-polypeptides from BChl *b*-containing species[33–36,43] (Fig. 5f), this could indicate that the *Rpi. globiformis* γ-like polypeptide is a new type of LH1 γ-polypeptide, perhaps one functionally best suited to the warm acidic conditions in which this unusual phototroph thrives.

## Discussion

Photosynthetic reaction centers bound with a multi-heme Cyt subunit are common in phototrophic purple bacteria (*Proteobacteria*)[1], and those with known structures all contain a truncated N-terminus and are modified by a lipid. By contrast, the structure of the LH1–RC from *Rpi. globiformis* (α1-subgroup in the *Proteobacteria*[44,45]) solved in the present work shows that its RC Cyt subunit is unique; it has an intact N-terminus that forms an α-helix in the photosynthetic membrane (Fig. 1e). This implies that a different strategy for membrane-anchoring of the Cyt subunit is necessary in this phototroph. Comparison of the N-terminal loop region (following the transmembrane helix) of the *Rpi. globiformis* Cyt with the C-terminal portion of PufX from *Rhodobacter* species (α3-subgroup) revealed remarkable structural similarities in terms of location within the complex and conformation including the topology of the transmembrane domain. Hence, our structural work here supports partly the hypothesis[10] that the phylogenetic roots of PufX lie in the N-terminus of the RC-bound Cyt subunit, furthermore suggesting that a common ancestor of phototrophic purple bacteria contained a full-length Cyt subunit that evolved independently in certain species through partial deletions of the *pufC* gene. Such deletions would have had at least two significant outcomes. First, the major portion of the Cyt subunit was cleft, generating an N-terminal fragment (corresponding to PufX) in *Rhodobacter* species[10] that evolved as a distinct protein to structurally organize the core complex. And second, in species lacking PufX, the major portion of the Cyt subunit that contains heme groups remained and was subsequently modified by an acyl lipid at its truncated Cys residue as a membrane anchor. These species include those with structures, such as *Blc. viridis*[12] (α2-subgroup), *Tch. tepidum*[13] (γ-*Proteobacteria*) and *Trv.* strain 970[14] (γ-*Proteobacteria*), and probably those with structures predicted from their sequences, such as *Rubrivivax* (*Rvi.*) *gelatinosus* (β-*Proteobacteria*) and *Alc. vinosum* (γ-*Proteobacteria*). Therefore, we hypothesize that the full-length *Rpi. globiformis* RC Cyt subunit represents a more ancient form than the lipid-anchored forms and more firmly associates with the RC to ensure efficient electron transfer.

The transmembrane domain of the *Rpi. globiformis* Cyt subunit is located close to the position occupied by protein-U in the *Rba. sphaeroides* LH1–RC (Fig. 3a), possibly as a consequence of spatial restriction. Because the interior of the LH1–RC is crowded, there is little space to accommodate a transmembrane domain, especially in closed-type LH1–RCs such as that of *Rpi. globiformis*. On the other hand, PufX is located on one edge of the open-shaped LH1 ring[18–23]. Occupation of this position by PufX prevents LH1 from forming a closed ring. This may be the reason why all known PufX-containing core complexes have an open LH1 ring. As a result, for the closed *Rpi. globiformis* LH1–RC that lacks protein-U, the position corresponding to protein-U and its immediate surroundings becomes the ideal candidate for accommodating the transmembrane domain of the *Rpi. globiformis* Cyt subunit. This is also applicable for the closed core complex of the filamentous green bacterium *Rof. castenholzii*. Similar to the *Rpi. globiformis*, *Rof. castenholzii* also lacks protein-U and its position is occupied by the N-terminal transmembrane domain of its Cyt subunit (Fig. 3e).

Inspection of the truncated N-terminal fragments of the Cyt subunits from *Trv.* strain 970, *Alc. tepidum* (Supplementary Fig. 10), *Blc. viridis* and *Tch. tepidum* (Supplementary Fig. 17a, b) showed hydrophobic signatures over a length of the membrane thickness, revealing their nature as transmembrane components. Modification of the Cys residues of these species by a lipid molecule at the truncated site coupled with the fact that the transmembrane domain of the *Rpi. globiformis* Cyt subunit is surrounded by phospholipid molecules at a similar position on the periplasmic surface (Fig. 2d) reflect a strong propensity of the

truncation site to associate with lipid molecules. The lipid-anchored Cyt subunits likely have a weaker association with the RC than that of *Rpi. globiformis* because some of them can be dissociated from the RC with detergent treatments as demonstrated in *Tch. tepidum*[46] and *Rvi. gelatinosus*[47].

Although no structural information was presented, an RC-bound multi-heme Cyt subunit was identified in the purified RC from the aerobic purple photosynthetic bacterium *Rsb. denitrificans* (α3-subgroup)[10,16], a species unable to carry out photosynthesis anaerobically. Resembling the *Rpi. globiformis* RC Cyt subunit, the subunit from this phototroph also has an intact N-terminus that was predicted from its sequence to form a transmembrane helix (Fig. 3f, Supplementary Fig. 17c). Similar cases have been inferred from sequence analyses of *Rhodovulum* sp.[25], *Chloroflexus* (*Cfl.*) *aurantiacus* (Cyt $c_{554}$)[2] and from actual structural studies of the Cyt subunit from *Rof. castenholzii*[3] (Fig. 3e); the latter two species are filamentous bacteria that emerge as the earliest phototrophs on a phylogenetic tree of 16 S rRNA genes[44].

Discovery of the γ-like polypeptides in the *Rpi. globiformis* LH1 was quite unexpected, since these have been found only in BChl *b*-containing purple phototrophs and used for distinguishing BChl *a*- and BChl *b*-containing species. The *Rpi. globiformis* γ-like polypeptide is the shortest among those reported, containing only 22 residues and showed relatively low sequence similarities to γ-polypeptides from BChl *b*-containing purple phototrophs (Fig. 5f). Its small size and rather distant relationship to γ-polypeptides from BChl *b*-containing species likely explain why the gene encoding the *Rpi. globiformis* γ-like polypeptide was not annotated as a hypothetical protein. This gene encoding the *Rpi. globiformis* γ-like polypeptide is located at a considerable distance from *pufLMC* gene clusters and is flanked by genes encoding a hypothetical protein and a methyl-accepting chemotaxis protein (Supplementary Fig. 5b). By contrast, genes encoding the *Blc. viridis* γ-polypeptides are clustered in a region flanked by two genes encoding hypothetical proteins. These differences in gene synteny between the two species further support the conclusion that the *Rpi. globiformis* γ-like polypeptide is a distinct form of LH1 γ-polypeptides. Based on their structural characteristics and positions in the LH1–RC, the *Rpi. globiformis* γ-like polypeptides likely play roles in both stabilizing the LH1 ring structure and regulating quinone transport between the RC and the acidic conditions of the quinone pool in the periplasm. In addition to the γ-like polypeptides, the presence of 3-methylhopanoids in *Rpi. globiformis*[48], lipids that help stabilize membranes and which are absent from most other anoxygenic phototrophs, is further evidence that anoxygenic photosynthesis in the warm acidic conditions of the *Rpi. globiformis* habitat may require special membrane components.

In summary, the structure of the *Rpi. globiformis* LH1–RC is a heretofore unseen version of this key photocomplex. The combination of an RC Cyt subunit anchored in the photosynthetic membrane by a polypeptide rather than an acyl lipid tail along with γ-like polypeptides and novel carotenoids makes the photocomplex of *Rpi. globiformis* structurally unique among all known purple bacteria. Moreover, along with the acidic, anoxic, and sulfidic warm-springs habitat of *Rpi. globiformis*—likely a common geological feature 3.4 billion years ago when photosynthesis first evolved—one could also speculate that the *Rpi. globiformis* LH1–RC contains the phylogenetic roots of PufX of *Rhodobacter* species and the γ-polypeptide of BChl *b*-containing purple bacterial species.

## Methods

**Preparation and characterization of the *Rpi. globiformis* LH1–RC complex.** Cells of *Rpi. globiformis* strain 7950 (DSM 161ᵀ) were cultivated phototrophically (anoxic/

light) at pH 5.1 and 25 °C for 7 days under incandescent light (60 W). Chromatophores were first treated with 0.30% (w/v) lauryldimethylamine N-oxide (LDAO) in 20 mM Tris-HCl buffer (pH 7.5) at 25 °C for 60 min to remove excess LH2, followed by centrifugation at 4 °C and 150,000 × g for 90 min. The pellets were then resuspended in the same buffer and extracted with 1.0% (w/v) n-dodecyl β-D-maltopyranoside (DDM) at 25 °C for 60 min followed by centrifugation at 4 °C and 150,000 × g for 90 min. The extracts were loaded onto sucrose gradient density tubes (five-stepwise concentrations: 10, 17.5, 25, 32.5, and 40% w/v) in 20 mM Tris-HCl buffer (pH 7.5) containing 0.05% w/v DDM followed by centrifugation at 4 °C and 150,000 × g for 6 h. The LH1–RC fractions were concentrated for absorption and circular dichroism (CD) measurements (Supplementary Fig. 1) and assessed by negative-stain EM using a JEM-1010 instrument (JEOL) (Supplementary Fig. 2a). Masses of the LH1 polypeptides were measured by matrix-assisted laser desorption/ionization time-of-flight mass spectroscopy (MALDI-TOF/MS)[49]. Carotenoids were analyzed by HPLC[50]. Suspension or solutions of membranes, LH1–RC and LH2 were directly injected into the HPLC system (Waters), which consisted of a μBondapak $C_{18}$ column and a photodiode array detector (Shimazdu). A linear gradient from methanol:water (9:1, v/v) to methanol was applied for 20 min and then methanol at flow-rate of 1.8 ml/min. Each carotenoid was also identified by their molecular masses (Supplementary Fig. 6). Quinones were extracted from chromatophores and analyzed using reverse-phase HPLC[51]. Phospholipids extracted from chromatophores were analyzed using $^{31}$P-NMR[52].

**Cryo-EM data collection**. Proteins for cryo-EM were concentrated to ~5 mg/ml. Then 2.5 microliters of the protein solution were applied on glow-discharged holey carbon grids (200 mesh Quantifoil R2/2 molybdenum) that had been treated with $H_2$ and $O_2$ mixtures in a Solarus II plasma cleaner (Gatan, Pleasanton, USA) for 30 s and then blotted, and plunged into liquid ethane at −182 °C using an EM GP2 plunger (Leica, Microsystems, Vienna, Austria). The applied parameters were a blotting time of 5 s at 80% humidity and 4 °C. Data were collected on a Titan Krios (Thermo Fisher Scientific, Hillsboro, USA) electron microscope at 300 kV equipped with a Falcon 3 camera (Thermo Fisher Scientific) (Supplementary Fig. 2b). Movies were recorded using EPU software (Thermo Fisher Scientific) at a nominal magnification of 96 k in counting mode and a pixel size of 0.820 Å at the specimen level with a dose rate of 0.69 e- per physical pixel per second, corresponding to 1.03 e- per Å$^2$ per second at the specimen level. The exposure time was 38.9 s, resulting in an accumulated dose of 40 e- per Å$^2$. Each movie includes 40 fractioned frames.

**Image processing**. All of the stacked frames were subjected to motion correction with MotionCor2[53]. Defocus was estimated using CTFFIND4[54]. A total of 421,464 particles were selected from 2,989 micrographs using the crYOLO[55] (Supplementary Fig. 3). All of the picked particles were further analyzed with RELION3.1[56], and 211,382 particles were selected by 2-D classification and divided into four classes by 3-D classification resulting in only one good class containing 128,119 particles. The initial 3-D model was generated in RELION. The 3-D auto refinement without any imposed symmetry (C1) produced a map at 2.35 Å resolution after contrast transfer function refinement, Bayesian polishing, masking, and post-processing. These particle projections were then subjected to subtraction of the detergent micelle density followed by 3-D auto refinement to yield the final map with a resolution of 2.24 Å according to the gold-standard Fourier shell correlation using a criterion of 0.143 (Supplementary Fig. 3b)[57]. The local resolution maps were calculated on RESMAP[58].

**Model building and refinement of the LH1–RC complex**. The initial atomic model for the LH1–RC structures was generated with MODELLER v10.1[59] using the atomic model of the *Tch. tepidum* LH1–RC (PDB: 5Y5S). The model was then fitted to the cryo-EM map obtained for the *Rpi. globiformis* LH1–RC using Chimera[60]. Amino acid substitutions and real-space refinement for the peptides and cofactors were performed using COOT[61]. Whole regions of the LH1 γ-like polypeptides as well as both terminal regions of the Cyt subunit and LH1 α-subunit were modeled ab-initio based on the density. The manually modified model was refined in real-space on PHENIX[62], and the COOT/PHENIX refinement was iterated until the refinements converged. Finally, the statistics calculated using MolProbity[63] were checked. The coordinates and restraints for carotenoid R.g. keto-II were generated using AceDRG[64] in the CCP4 suite[65]. Figures were drawn with the Pymol Molecular Graphic System (Ver2.5, Schrödinger)[66] and UCSF Chimera[60]. Surface charge distributions were calculated using the APBS Electrostatics plugin in the Pymol.

**Reporting summary**. Further information on research design is available in the Nature Portfolio Reporting Summary linked to this article.

## Data availability
Map and model have been deposited in the EMDB and PDB with the accession codes: EMD-33501 and PDB-7XXF. All other data are available from the authors upon reasonable request.

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

## Acknowledgements

We thank C. Tanaka for providing technical assistance. This research was partially supported by Platform Project for Supporting Drug Discovery and Life Science Research (Basis for Supporting Innovative Drug Discovery and Life Science Research (BINDS) from AMED under Grant Numbers JP20am0101118 (support number 1758) and JP20am0101116 (support number 1878), 17am0101116j0001, 18am0101116j0002, 19am0101116j0003, and 22am121004. R.K., M.H., and B.M.H. acknowledge the generous support of the Okinawa Institute of Science and Technology (OIST), Scientific Computing & Data Analysis Section at OIST and the Japanese Cabinet Office. L.-J.Y. acknowledges the financial supports from the National Key R&D Program of China (2021YFA0909600). This work was supported in part by JSPS KAKENHI Grant Numbers JP16H04174, JP18H05153, JP20H05086, and JP20H02856, Takeda Science Foundation, and the Kurata Memorial Hitachi Science and Technology Foundation, Japan.

## Author contributions

Z.-Y.W.-O. and K.T. designed the work, M.T.M. provided materials, K.T., R. K., K.K., S.T., and M.H. performed the experiments, K.T., R.K., K.V.P.N., L.-J.Y., Y.K., M.T.M., A.M., B.M.H., and Z.-Y.W.-O. analyzed data, Z.-Y.W.-O., K.T., and M.T.M. wrote the manuscript.

## Competing interests

The authors declare no competing interests.
