## [Peer Review File · Communications Biology]

Reviewers' comments:

Reviewer #1 (Remarks to the Author):

In this work, Tani et al. reported a cryo-EM structure of the RC-LH1 complex from *Rhodospila globiformis*, an acidophilic anoxygenic phototrophic purple nonsulfur bacterium. Their results showed that the RC-LH1 structure involves a full-length cytochrome subunit and γ -like polypeptides in the LH1 assemblies. The Cyt subunit contains a C-terminal domain that associates with the RC and an N-terminal region that forms a transmembrane helix within RC-LH1. Part of the N-terminal region shares similar locations with *Rhodobacter* PufX, suggesting thus a possible evolutionary relationship between Cyt and PufX. The extra γ -like polypeptide forms α -helix and interacts with the LH1 β -polypeptide transmembrane region, reminiscent of the γ polypeptide recently reported in the BChl b-containing purple photosynthetic bacterium *Blc. viridis* (Nature 2018, 556: 203-208). Collectively, the authors proposed the potential evolutionary status of *Rpi. globiformis* between BChl a-containing and BChl b-containing species. The study shows some interesting features of the *Rpi. globiformis* RC-LH1, and the manuscript and results are clearly presented in general. However, there are some concerns that should be addressed.

Major comments:

Previous study has reported a RC-LH1 structure from *R. castenholzii*, which contains a full-length Cyt C subunit including the N-terminal transmembrane domain. It is inappropriate to claim that the present manuscript represents the first study on a full Cyt subunit. The authors should tune down the novelty of their findings.

It is not convincing that *Rhodobacter* PufX and *Rpi. globiformis* Cyt N-terminus have a "striking" close relationship in terms of their positions and location. The sequence alignment of PufX and *R. castenholzii* Cyt N-terminus does not show high similarity (even the loop region) and the locations of their transmembrane regions are very different (Fig. 3a, 3f). In addition, Fig. 3f should show a proper sequence alignment is required.

Instead, the N-terminal transmembrane region shares similar locations with a newly identified "Protein-U" in *Rba. sphaeroides* RC-LH1 structures, as shown in Fig. 3a. However, "Protein-U" and its relationship with Cyt have not been well discussed in the manuscript.

Only 11 γ -like polypeptides were identified within 16 LH1 pairs. As the peripheral γ -like polypeptides showed higher flexibility and relatively lower densities within the RC-LH1 (Supplementary Fig. 3d), how are the authors confident with the number and positions of γ -like polypeptides per RC-LH1? It is useful to provide quantitative analysis of LH1 α , β , and γ -like polypeptides, for example based on mass spectrometry, and present in more detail how the number and asymmetric locations of γ -like polypeptides were determined precisely. Are the locations in the absence of γ -like polypeptides random or specific among RC-LH1 structures? It is also useful to show local densities at the positions where there are no γ -like peptides.

Phylogenetic analysis of *Rpi. globiformis* and other purple photosynthetic bacterial species, particularly the species possessing PufX, full/partial Cyt subunits, and γ (-like) polypeptides, should be provided, in support of the implication of *Rpi. globiformis*' unique evolutionary status.

Other comments:

In Abstract: the general information of *pufC* should be provided.

Fig. 1: It is important to show the cryo-EM map of the RC-LH1 complex.

Recent studies on the *R. sphaeroides* RC-LH1 monomer and dimer which both contain PufX should be acknowledged, in order to provide an overview of the research background to readers (Nature Communications, 2022, 13: 1904; Biochemical Journal, 2021, 478: 3923-3937; Nature Communications, 2022, 13: 1977).

Line 138: specify what is unique.

The interactions between γ -like polypeptides and LH1 β subunits in *Rpi. globiformis* and *Blc. viridis* should be compared and discussed.

Supplementary Fig. 1: What does the major protein band above the band of Cyt C in SDS-PAGE represent?

Supplementary Fig. 4: It is useful to show the electron densities of entire Cyt subunit.

Supplementary Fig. 8: Add the units of electrostatic potential.

Reviewer #2 (Remarks to the Author):

In this work, the authors determined a 2.24 Å-resolution cryo-EM structure of the light-harvesting 1-reaction center (LH1-RC) complex from *Rpi. Globiformis*, the most acidophilic anaerobic purple bacteria, and characterized the unique features through extensive structural comparisons with LH1-RCs from other photosynthetic bacteria. Different from the known structure of purple bacterial LH1-RC, *Rpi. Globiformis* LH1-RC contains a RC-bound full-length tetraheme cyt c subunit, which was embedded in the membrane through an intact N-terminal transmembrane helix. In addition, the structure contains eleven copies of γ -like peptide located between the β -polypeptides in the LH1 ring, which has previously been found only in the BChl-b containing bacterium *Blc. viridis*. These γ -like polypeptides block the pores formed between LH1 $\alpha\beta$ pairs that are presumably function as quinone transport channels. Especially, blockage of one other channel by an additional UQ and a carotenoid near the QA site implies a mechanism for quinone exchange only through the QB site. This work is of great interest to readers in the field of prokaryotic photosynthesis. *Rpi. Globiformis* LH1-RC contains chimeric structural features of bacterial LH-RC complexes. The conformation of the RC-bound tetraheme cyt c subunit resembles that of the LH-RC from *Roseiflexus castenholzii*, and the additional γ -like peptides adopted same locations and orientations as that in the LH1-RC of *Blc. Viridis*. The structural features are professionally presented, and the conclusions of this paper are mostly well supported by the data. Overall, the results of this work will contribute to broaden the structural diversity of bacterial LH1-RC complexes, and are worthy to be published.

However, the English writing needs to be modified to concisely present the work. And few questions need to be clarified and extended:

1.Line180-186, the authors compared the structure of *Rpi. Globiformis* LH1-RC with *Rba. sphaeroides* monomeric LH1-RC, and found that "the C-terminus (Leu55-Gly69) of PufX is indeed closely located and parallel to a loop region (Phe27-Arg40) of the Cyt subunit immediately following its N-terminal transmembrane domain". However, the transmembrane helices of PufX and Cyt c are not aligned at all. In addition, the PufX and cyt c play different roles in the LH1-RC complex of purple bacteria. It has been reported by several studies that PufX plays a critical role in initiating the assembly and mediating the dimerization of LH1-RC, whereas the cyt c subunit mainly functions in mediating the electron transfer. Therefore, it is not easy to understand the statement that "PufX from *Rhodobacter* species may have had its structural origin in the intact N terminus of the Cyt subunit", which needs to be supported by even more stronger evidences.

2.Line 85-86, "pufX is positioned in the puf operon at the same location as pufC in the genome of species with an RC-bound multi-heme Cyt subunit—immediately following pufM,". Please rewrite this sentence to make it clear and understandable.

3.Line 194, *Rpi. Globiformis* LH1-RC and *Roseiflexus (Rof.) castenholzii* RC-LH both contain a tetraheme cyt c subunit that was tightly bound with the RC through an intact N-terminal alpha helix. Considering the novelty of *Rpi. globiformis* Cyt subunit, it is necessary to compare the cyt c subunits from these two structures in Figure 3e, instead of comparing the structures of *Rba. sphaeroides* monomeric LH1-RC and *Roseiflexus (Rof.) castenholzii* LH-RC. The comparisons between PufX and cyt c of *Roseiflexus (Rof.) castenholzii* LH-RC should be moved to the supplementary files.

4.Line 212-214, it is not accurate to state that "the affinity between the *Rpi. globiformis* Cyt subunit and LH1-RC is stronger than that in *Trv. strain 970* and *Alc. tepidum*." merely based on the surface contact area of these two subunits. It is better to only describe that "*Rpi. globiformis* Cyt subunit contains larger surface contact area with the LH1-RC than the other bacterial LH1-RC".

5.Line 219-220, these specific structural features of the *Rpi. globiformis* LH1-RC may be necessary to

stabilize the binding of Cyt subunit with the RC, but they are not related to the stability of the complex under acidic pH. Please revise this kind of statements all through the manuscript.

Reviewer #3 (Remarks to the Author):

The manuscript by Tani and coworkers describes a structural study of the proteins that form the photocomplex from *Rpi globiformis*. This membrane bound complex captures light energy and performs the initial photochemistry. A three-dimensional model is determined using cryo-EM data obtained from purified protein and appears to have been carefully analyzed, producing a high-quality structure. In the past few years, there have been many cryo-EM structures of the light-harvesting-reaction center complex. Overall, the bulk of the model has the same key structural features of the previously reported models, including the presence of a ring of light-harvesting proteins surrounding the central reaction center, as shown in Figure 1. However, this model has the distinguishing aspect not previously evident, namely the presence of a transmembrane helix at the N-terminus of the tetraheme cytochrome subunit. Bound cytochromes have been well characterized but the structures lacked the helix due to processing. The interactions of the cytochrome are described with a focus on the N-terminal region (Figures 2-4). The model also has a resolved PufX subunit and a gamma-like subunit. The structural discussion of the N-terminus region of the tetraheme cytochrome should be of interest to the journal's audience, as it has implications for the roles of small proteins in the assembly and stability of large membrane protein complexes. I recommend acceptance, pending a few small modifications.

Minor points: The title is misleading as the complex is actually very similar to many other complexes, it is only the structurally small feature of the N-terminus of the cytochrome that is unique. The language concerning the genetic implications is very loose and should be removed. The presented data shows a structural relationship between the N-terminus and PufX but nothing supporting the "missing link" concept highlighted in the abstract and discussion. A careful genetic analysis of the genes from many different organisms would be required to support this idea. The authors should simply note the structural correlation.

Response to reviewers:

Reviewer #1

Reviewer #1's comments: Point 1

In this work, Tani et al. reported a cryo-EM structure of the RC-LH1 complex from *Rhodospira globiformis*, an acidophilic anoxygenic phototrophic purple nonsulfur bacterium. Their results showed that the RC-LH1 structure involves a full-length cytochrome subunit and γ -like polypeptides in the LH1 assemblies. The Cyt subunit contains a C-terminal domain that associates with the RC and an N-terminal region that forms a transmembrane helix within RC-LH1. Part of the N-terminal region shares similar locations with *Rhodobacter* PufX, suggesting thus a possible evolutionary relationship between Cyt and PufX. The extra γ -like polypeptide forms α -helix and interacts with the LH1 β -polypeptide transmembrane region, reminiscent of the γ polypeptide recently reported in the BChl b-containing purple photosynthetic bacterium *Blebsphaera viridis* (Nature 2018, 556: 203-208). Collectively, the authors proposed the potential evolutionary status of *Rpi. globiformis* between BChl a-containing and BChl b-containing species. The study shows some interesting features of the *Rpi. globiformis* RC-LH1, and the manuscript and results are clearly presented in general. However, there are some concerns that should be addressed.

Major comments:

Previous study has reported a RC-LH1 structure from *R. castenholzii*, which contains a full-length Cyt C subunit including the N-terminal transmembrane domain. It is inappropriate to claim that the present manuscript represents the first study on a full Cyt subunit. The authors should tune down the novelty of their findings.

Our response:

We thank the reviewer's positive assessment of our work. *Rof. castenholzii* is a filamentous anoxygenic phototroph (previously called green nonsulfur bacterium) that is phylogenetically distant from other anoxygenic photosynthetic bacteria. We described our structure of *Rpi. globiformis* LH1-RC as the first one in a purple bacterium that has an intact RC-bound Cyt with a full N-terminus. However, according to the reviewer's suggestion and in order to avoid misunderstanding, we have changed the statements on this point.

Reviewer #1's comments: Point 2

It is not convincing that *Rhodobacter* PufX and *Rpi. globiformis* Cyt N-terminus have a "striking" close relationship in terms of their positions and location. The sequence alignment of PufX and *R. castenholzii* Cyt N-terminus does not show high similarity (even the loop region) and the locations of their transmembrane regions are very different (Fig. 3a, 3f). In addition, Fig. 3f should show a proper sequence alignment is required.

Instead, the N-terminal transmembrane region shares similar locations with a newly identified "Protein-U" in *Rba. sphaeroides* RC-LH1 structures, as shown in Fig. 3a. However, "Protein-U" and its relationship with Cyt have not been well discussed in the manuscript.

Our response:

- When we compared the positions and structural conformations of the *Rba. sphaeroides* PufX and the N-terminus of *Rpi. globiformis* Cyt subunit, rather than just their sequences, we concluded that the similarities were indeed remarkable. And this is also true if we compare the sequences of the overlapping transmembrane regions (Supplementary Fig. 9b), as well, although the overall sequence similarity is not very high. Regarding the *Rof. castenholzii*, the location of its transmembrane region is very different from that of PufX, as pointed out by the

reviewer; however, the topologies of their overall structures of the N-terminal domains are quite similar (Fig. 3e). The possible reasons for the difference in their locations are listed in the next paragraph and may also be due to the evolutionary (phylogenetic) differences between the two species as mentioned above (Point 1). However, according to the reviewer's suggestion and in order to avoid misunderstanding, we have modified the statements on these points.

- The transmembrane domain of *Rpi. globiformis* Cyt subunit is located close to the protein-U in *Rba. sphaeroides* LH1-RC (Fig. 3a). This is likely a consequence of spatial restriction. Because the interior of LH1-RC is crowded, there is little space to accommodate a transmembrane domain, especially for the closed-type LH1-RC such as that of *Rpi. globiformis*. On the other hand, PufX is located on one edge of the open-shaped LH1 ring. Occupation of this position by PufX prevents the LH1 from forming a closed ring. This may be the reason why all known PufX-containing core complexes have an open LH1 ring. As a result, for the *Rpi. globiformis* LH1-RC that has a closed LH1 ring and lacks protein-U, the position corresponding to that of protein-U becomes one of the best candidates for accommodating the transmembrane domain of *Rpi. globiformis* Cyt subunit. This is also seen in the *Rof. castenholzii* core complex (Fig. 3e) that has a closed LH1 ring and lacks protein-U where the position is instead occupied by the N-terminal domain of Cyt subunit. According to the reviewer's suggestion, we have added discussions on this point.
- The sequences in Fig. 3f have been re-aligned to match the loop regions (green fonts) of the N-termini of Cyt subunits to the C-terminus of the PufX (orange fonts) according to the reviewer's suggestion.

Reviewer #1's comments: Point 3

Only 11 γ -like polypeptides were identified within 16 LH1 pairs. As the peripheral γ -like polypeptides showed higher flexibility and relatively lower densities within the RC-LH1 (Supplementary Fig. 3d), how are the authors confident with the number and positions of γ -like polypeptides per RC-LH1? It is useful to provide quantitative analysis of LH1 α , β , and γ -like polypeptides, for example based on mass spectrometry, and present in more detail how the number and asymmetric locations of γ -like polypeptides were determined precisely. Are the locations in the absence of γ -like polypeptides random or specific among RC-LH1 structures? It is also useful to show local densities at the positions where there are no γ -like peptides.

Our response:

- Discovery of the γ -like polypeptides in the LH1 complex of a BChl *a*-containing purple bacterium was quite surprising. At such high resolution of 2.24 Å, we are confident of the number and positions of these polypeptides because we can unambiguously extract most sidechains of the polypeptide from the density map. This enabled us to derive its amino acid sequence as an unannotated protein from the genome, and can easily distinguish between a polypeptide and a lipid (or detergent) molecule. To demonstrate this and to answer the reviewer's questions, we have added three structures (Fig. 1c, Fig. 5b and in Supplementary Fig. 4b) and a new Supplementary Fig. 14. A total of 11 γ -like polypeptides were identified at the positions between the peripheral LH1 β - β pairs while the remaining 5 corresponding sites were occupied by cardiolipins (Fig. 1b, 1d and Fig. 5b). Inspection of the asymmetrically distributed binding sites of the γ -like polypeptides revealed a high structural homogeneity in their binding environments with nearby β -polypeptides (Supplementary Fig. 14b, *left*), whereas the corresponding cardiolipin-binding sites showed relatively large structural inhomogeneities (Supplementary Fig. 14b, *right*) indicating specific bindings of the γ -like polypeptide and cardiolipin to these sites. These structural features may reflect an overall structural heterogeneity of the large LH1-RC complex in a dynamically flexible membrane environment in which the LH1-RCs constantly undergo a "breathing motion" as reported in the literature (e.g. *J. Biol. Chem.*, **279**, 21327, 2004). The number and unique arrangement of the γ -like polypeptides may also serve as a mechanism for maintaining an adequate number

of pathways for the quinone transport. According to the reviewer's suggestion, we have added descriptions on this issue (p. 11–12).

- A quantitative analysis of the LH1 α , β , and γ -like polypeptides by biochemical and/or spectroscopic methods would not be easy (if indeed it is even possible) and far less definitive (with large experimental errors) than the direct structure determination at atomic resolutions (this work). For example, the MALDI-TOF/MS spectroscopy in Supplementary Fig. 13 can only show the m/z values but not the molar ratios of these polypeptides because the peak intensities are heavily dependent on the quality of the co-crystals formed between these polypeptides and the matrix; i. e., the peak intensities are typically different between α - and β -polypeptides even though they are equal in number. The quality of the co-crystals is highly dependent on the properties (hydrophobicity, chain length, etc) of the membrane polypeptides. Therefore, there is no direct correlation between the signals in mass spectroscopy and molar ratio of the polypeptides. On the other hand, the γ -like polypeptide is too short (22 aa) to be clearly resolved in SDS-PAGE for a reliable quantitative analysis.

Reviewer #1's comments: Point 4

Phylogenetic analysis of *Rpi. globiformis* and other purple photosynthetic bacterial species, particularly the species possessing PufX, full/partial Cyt subunits, and γ (-like) polypeptides, should be provided, in support of the implication of *Rpi. globiformis*' unique evolutionary status.

Our response:

- We have prepared two sets of phylogenetic analysis. Fig. R1 (below) shows such an alignment along with a phylogenetic tree based on the amino acid sequences of PufX and the N-terminal regions of PufC. The phylogenetic analysis revealed close relationships of the *Rpi. globiformis* PufC-N-terminus with that of *Rvi. gelatinosus*, followed by those of *Blc. viridis* and purple sulfur bacteria and by *Rhodobacter* PufX. The PufC-N-terminus of *Rof. castenholzii* is located at a relatively distant position in the phylogenetic tree. However, it is difficult to make a more meaningful comparison because these sequences are too short. Therefore, we prefer this figure to be used for review only.

Fig. R1 Phylogenetic trees based on the amino acid sequences of PufX and N-terminal regions of PufC. Sequences were aligned using the ClustalX program and gaps were corrected by manual inspection.

The phylogenetic tree was drawn using the MEGA ver. 3.1 program. All gaps in the sequence alignment were omitted in a pairwise manner. Construction of the trees was performed by the neighbor-joining method, applying the *p*-distance as a distance estimator. *Rof. castenholzii* PufC was used as an outgroup. Bootstrap values are presented at the corresponding nodes.

- Because the γ (-like) polypeptides are also too short to draw a meaningful conclusion, we have prepared a phylogenetic tree based on 16S rRNA analyses among *Rpi. globiformis* and other purple bacteria (Fig. R2, below). This is somewhat similar to that for the classification of Proteobacteria by Woese (*Microbiol. Rev.* **51**, 221, 1987). It is of interest to note that *Rpi. globiformis* (α 1-subgroup) is a close relative of the acidophilic bacterium *Acidiphilium rubrum* (α 1-subgroup), but is clearly distant from *Rhodobacter* species (α 3-subgroup). Because the 16S rRNA analysis has been reported previously and reflects an evolutionary characteristic of the purple bacteria rather than a portion of a specific protein, we prefer this figure to be used for review only. Instead, we have added the phylogenetic subgroup names of Proteobacteria for the relevant bacteria in text of the revised manuscript.

Fig. R2 16S rRNA analysis. The phylogenetic tree was drawn using the programs ClustalX and MEGA ver. 3.1. All gaps in the sequence alignment were omitted, and tree constructed by the neighbor-joining method, applying the Kimura 2-parameter distance as a distance estimator. Both transitional and transversional replacements were taken into account.

Reviewer #1's comments: Other comments

In Abstract: the general information of *pufC* should be provided.

Fig. 1: It is important to show the cryo-EM map of the RC-LH1 complex.

Recent studies on the *R. sphaeroides* RC-LH1 monomer and dimer which both contain PufX should be acknowledged, in order to provide an overview of the research background to readers (Nature Communications, 2022, 13: 1904; Biochemical Journal, 2021, 478: 3923-3937; Nature Communications, 2022, 13: 1977).

Line 138: specify what is unique.

The interactions between γ -like polypeptides and LH1 β subunits in *Rpi. globiformis* and *Blc. viridis* should be compared and discussed.

Supplementary Fig. 1: What does the major protein band above the band of Cyt C in SDS-PAGE represent?

Supplementary Fig. 4: It is useful to show the electron densities of entire Cyt subunit.

Supplementary Fig. 8: Add the units of electrostatic potential

Our response:

- General information about pufC has been provided in Abstract.
- We have added a cryo-EM map of the LH1–RC complex in Fig. 1c.
- Literature on the *Rba. sphaeroides* LH1–RC has been added.
- The definition of “unique” is “one of a kind” or “being the only one of a particular type”. The word “unique” was thus used to describe *Rpi. globiformis* in general and its LH1–RC in particular because: (1) *Rpi. globiformis* has a distinct phylogeny, (2) the keto-carotenoids of *Rpi. globiformis* LH1 have only been found in this phototroph (as discussed in the literature Ref. 27 and 37), (3) No other anaerobic purple bacterium has been shown to have a full length LH1–RC Cyt subunit, and (4) No LH1–RC from a bacteriochlorophyll *a*-containing purple bacterium whose structure is known contains γ -like polypeptides. We present these criteria more clearly in the revised manuscript to justify our usage of the word “unique”.
- We have added a description on the interactions between γ -like polypeptides and LH1 β subunits in *Rpi. globiformis* and *Blc. viridis* as suggested by the reviewer.
- We do not know the exact identity of the major protein band above the Cyt subunit in SDS-PAGE. It might be residual aggregates of the incompletely denatured RC proteins based on its molecular weight.
- We have added electron densities of the entire Cyt subunit in Supplementary Fig. 4b.
- The units of electrostatic potential have been added in Supplementary Fig. 8.

Reviewer #2

Reviewer #2's comments: Point 1

In this work, the authors determined a 2.24 Å-resolution cryo-EM structure of the light-harvesting 1–reaction center (LH1–RC) complex from *Rpi. Globiformis*, the most acidophilic anaerobic purple bacteria, and characterized the unique features through extensive structural comparisons with LH1–RCs from other photosynthetic bacteria. Different from the known structure of purple bacterial LH1–RC, *Rpi. Globiformis* LH1–RC contains a RC-bound full-length tetraheme cyt c subunit, which was embedded in the membrane through an intact N-terminal transmembrane helix. In addition, the structure contains eleven copies of γ -like peptide located between the β -polypeptides in the LH1 ring, which has previously been found only in the BChl-b containing bacterium *Blc. viridis*. These γ -like

polypeptides block the pores formed between LH1 $\alpha\beta$ pairs that are presumably function as quinone transport channels. Especially, blockage of one other channel by an additional UQ and a carotenoid near the QA site implies a mechanism for quinone exchange only through the QB site.

This work is of great interest to readers in the field of prokaryotic photosynthesis. *Rpi. Globiformis* LH1-RC contains chimeric structural features of bacterial LH-RC complexes. The conformation of the RC-bound tetraheme cyt *c* subunit resembles that of the LH-RC from *Roseiflexus castenholzii*, and the additional γ -like peptides adopted same locations and orientations as that in the LH1-RC of *Blc. Viridis*. The structural features are professionally presented, and the conclusions of this paper are mostly well supported by the data. Overall, the results of this work will contribute to broaden the structural diversity of bacterial LH1-RC complexes, and are worthy to be published.

However, the English writing needs to be modified to concisely present the work. And few questions need to be clarified and extended:

1.Line180-186, the authors compared the structure of *Rpi. Globiformis* LH1-RC with *Rba. sphaeroides* monomeric LH1-RC, and found that “the C-terminus (Leu55–Gly69) of PufX is indeed closely located and parallel to a loop region (Phe27–Arg40) of the Cyt subunit immediately following its N-terminal transmembrane domain”. However, the transmembrane helices of PufX and Cyt *c* are not aligned at all. In addition, the PufX and cyt *c* play different roles in the LH1-RC complex of purple bacteria. It has been reported by several studies that PufX plays a critical role in initiating the assembly and mediating the dimerization of LH1-RC, whereas the cyt *c* subunit mainly functions in mediating the electron transfer. Therefore, it is not easy to understand the statement that “PufX from *Rhodobacter* species may have had its structural origin in the intact N terminus of the Cyt subunit”, which needs to be supported by even more stronger evidences.

Our response:

- We appreciate the reviewer’s positive assessment of our work. The fact that the transmembrane helices of PufX and the Cyt subunit are not at the same position is thought to be a consequence of spatial restriction. Because the interior of LH1–RC is crowded, there is little space to accommodate a transmembrane domain, especially in such a “closed-type” LH1–RC such as that of *Rpi. globiformis*. On the other hand, PufX is located on one edge of the open-shaped LH1 ring. Occupation of this position by PufX prevents the LH1 from forming a closed ring. This may be why all known PufX-containing core complexes have an open LH1 ring. As a result, for the *Rpi. globiformis* LH1–RC that has a closed LH1 ring, the transmembrane helix of the Cyt subunit is forced to locate at a different position from that of PufX. The location of the transmembrane helix of the *Rpi. globiformis* Cyt subunit is actually close to that of *Rba. sphaeroides* protein-U (Fig. 3a, 3b), a protein absent from *Rpi. globiformis*, therefore this location becomes one of the best candidates for accommodating the transmembrane domain of the *Rpi. globiformis* Cyt subunit. This is also seen in the *Rof. castenholzii* core complex (Fig. 3e); this complex has a closed LH1 ring and lacks protein-U where the position is occupied instead by the N-terminal domain of the Cyt subunit. Additionally, a shorter polypeptide (X in Fig. 3e) close to PufX also adopted a different orientation from that of PufX. We have added discussions on this point in the revised manuscript.
- In this work, we argue that the full-length *Rpi. globiformis* RC Cyt subunit represents a more ancient form than its truncated forms as seen in the lipid-anchored Cyt subunit of *Blc. viridis* and the PufX in *Rhodobacter* species. The functional domain containing tetraheme binding sites in all RC Cyt subunits is in the soluble portion protruded on the periplasmic side, whereas the transmembrane domain of the Cyt subunit plays a structural (different) role in anchoring the functional portion to the membrane forming a more firmly association with the RC to ensure efficient electron transfer. Both the lipid-anchored Cyt subunit and PufX are thought to be an evolutionary consequence of cleavage of the full-length *pufC* gene that resulted in two separate proteins: one (lipid-anchored Cyt subunit) still retains its functional role as an electron transfer component; and another (PufX) evolved to specialize in structural

stabilization and assembly of the LH1–RC. We have tried diligently to show and to discuss throughout the manuscript the fact that PufX has many structural and topological features similar to those of the intact N-terminus of the Cyt subunit based on both sequence alignments and conformational comparisons.

Reviewer #2's comments: Point 2

2. Line 85-86, “pufX is positioned in the puf operon at the same location as pufC in the genome of species with an RC-bound multi-heme Cyt subunit—immediately following pufM,”. Please rewrite this sentence to make it clear and understandable

Our response:

We have modified this sentence.

Reviewer #2's comments: Point 3

3. Line 194, *Rpi. Globiformis* LH1-RC and *Roseiflexus (Rof.) castenholzii* RC-LH both contain a tetraheme cyt c subunit that was tightly bound with the RC through an intact N-terminal alpha helix. Considering the novelty of *Rpi. globiformis* Cyt subunit, it is necessary to compare the cyt c subunits from these two structures in Figure 3e, instead of comparing the structures of *Rba. sphaeroides* monomeric LH1–RC and *Roseiflexus (Rof.) castenholzii* LH-RC. The comparisons between PufX and cyt c of *Roseiflexus (Rof.) castenholzii* LH-RC should be moved to the supplementary files.

Our response:

Because Fig. 3 was intended to focus on the comparison between PufX and N-terminal portions of the Cyt subunits from various bacteria and the fact that *Rof. castenholzii* is the only species that has a full-length Cyt subunit with known structure, we prefer to retain Fig. 3e as it is. However, to address the reviewer's concern, we have added structural comparisons of the Cyt subunits between *Rpi. globiformis* and *Rof. castenholzii* in Fig. 2c (Fig. 2 is focused on the structure of the *Rpi. globiformis* Cyt subunit).

Reviewer #2's comments: Point 4

4. Line 212-214, it is not accurate to state that “the affinity between the *Rpi. globiformis* Cyt subunit and LH1–RC is stronger than that in *Trv. strain 970* and *Alc. tepidum*.” merely based on the surface contact area of these two subunits. It is better to only describe that “*Rpi. globiformis* Cyt subunit contains larger surface contact area with the LH1-RC than the other bacterial LH1-RC”.

Our response:

We have modified this description according to the reviewer's suggestion.

Reviewer #2's comments: Point 5

5. Line 219-220, these specific structural features of the *Rpi. globiformis* LH1–RC may be necessary to stabilize the binding of Cyt subunit with the RC, but they are not related to the stability of the complex under acidic pH. Please revise this kind of statements all through the manuscript.

Our response:

We have removed the description according to the reviewer's suggestion.

Reviewer #3

Reviewer #3's comments

The manuscript by Tani and coworkers describes a structural study of the proteins that form the photocomplex from *Rpi globiformis*. This membrane bound complex captures light energy and performs the initial photochemistry. A three-dimensional model is determined using cryo-EM data obtained from purified protein and appears to have been carefully analyzed, producing a high-quality structure. In the past few years, there have been many cryo-EM structures of the light-harvesting-reaction center complex. Overall, the bulk of the model has the same key structural features of the previously reported models, including the presence of a ring of light-harvesting proteins surrounding the central reaction center, as shown in Figure 1. However, this model has the distinguishing aspect not previously evident, namely the presence of a transmembrane helix at the N-terminus of the tetraheme cytochrome subunit. Bound cytochromes have been well characterized but the structures lacked the helix due to processing. The interactions of the cytochrome are described with a focus on the N-terminal region (Figures 2-4). The model also has a resolved PufX subunit and a gamma-like subunit. The structural discussion of the N-terminus region of the tetraheme cytochrome should be of interest to the journal's audience, as it has implications for the roles of small proteins in the assembly and stability of large membrane protein complexes. I recommend acceptance, pending a few small modifications.

Minor points: The title is misleading as the complex is actually very similar to many other complexes, it is only the structurally small feature of the N-terminus of the cytochrome that is unique. The language concerning the genetic implications is very loose and should be removed. The presented data shows a structural relationship between the N-terminus and PufX but nothing supporting the "missing link" concept highlighted in the abstract and discussion. A careful genetic analysis of the genes from many different organisms would be required to support this idea. The authors should simply note the structural correlation.

Our response:

- We thank the reviewer's positive assessment of our work. The title we have chosen is a reflection of more than just the fact that the *Rpi. globiformis* LH1-RC has a full-length Cyt subunit. The complex also contains 11 copies of a protein (the γ -like polypeptides) absent from the LH1-RC complexes of all purple bacteria containing bacteriochlorophyll *a* whose structures are known. Moreover, the keto-carotenoids present in the complex are also absent from all other anoxygenic phototrophs. Collectively then, we feel that the word "unique", as defined above in response to Reviewer #1's other comments, is a correct and appropriate descriptor of the *Rpi. globiformis* LH1-RC; the structure is indeed "one-of-a-kind". We have worked this theme more convincingly into the abstract and text of the revised manuscript.
- Our work aims at unravelling the structural and phylogenetic relationships between the N-terminus of RC Cyt subunit and PufX, and we have tried our best to show and to discuss this issue objectively. However, in order to address the reviewer's concern, we have added subgroup names of the Proteobacteria for the relevant species to strengthen our comparisons between these two proteins within the Proteobacteria.

REVIEWERS' COMMENTS:

Reviewer #1 (Remarks to the Author):

The authors have addressed most of the comments.
To improve the readability, it would be useful to add the *Rpi. globiformis* PufC protein sequence information into the manuscript.

Reviewer #2 (Remarks to the Author):

The authors have carefully addressed all the questions raised in the first revision, including the evolutionary and phylogenic analyses of the PufX and RC-bound cyt c subunits in purple and green non-sulfur bacteria, and also the structural and functional features of the γ -like peptides in LH1. Considering the structural novelty, and the insight it provides into the diversity and evolution of bacterial photosynthetic complexes, I recommend acceptance of this manuscript.

Reviewer #3 (Remarks to the Author):

The authors have carefully addressed the reviewer comments. I recommend publication of the revised manuscript.